# Wolfhart Pannenberg's Theological Method and Metaphysics

Kyungrae Kim

Systematic Theology, Presbyterian University and Theological Seminary, Seoul 04965, Republic of Korea; dkr978@puts.ac.kr

**Abstract:** After the rise of logical positivism, even in the realm of theology, there was a trend to give up on accepting the actions of God in history as objective acts and to create an atmosphere of separating faith and reason. Wolfhart Pannenberg's work presents a compelling integration of theology with the rational and empirical rigors of the scientific age. Through a comprehensive theological method, he aimed to establish a dialogue between faith and scientific inquiry, challenging the exclusivity of logical positivism by proposing a theological metaphysics grounded in the concept of retroactive ontology. Pannenberg's approach is distinguished by its systematic application of hermeneutics, considering the totality of history as the context for divine revelation, and positioning the resurrection of Jesus Christ as a pivotal event that embodies God's influence on the world. His innovative ontology, which enables one to consider divine action as objective, seeks to validate theology as a science, engaging with natural sciences to foster a mutual enrichment between faith and reason. Pannenberg's methodological rigor and metaphysical framework offer a robust foundation for a theology that is both intellectually defensible and deeply rooted in Christian faith, advocating for a theology of nature that reconciles the divine with the empirical world.

**Keywords:** metaphysics; Wolfhart Pannenberg; method; natural theology; theology of nature

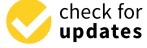



## 1. Introduction

Wolfhart Pannenberg studied broadly in many areas, including the natural sciences, and sought to apply the concepts of these sciences to theology; interpret them from a theological perspective; and then offer these theological reflections back to other sciences. His work sought to establish a theology through which the pursuit of divine understanding could withstand the scrutiny of rational inquiry and empirical evidence. Through his efforts, Pannenberg not only preserved the academic integrity of theology but also paved the way for its dynamic engagement with the philosophical and scientific advancements of the scientific age.

In the context where logical positivism was spreading and only verifiable things were considered academically meaningful statements, a new metaphysics was needed to discuss the existence of God and His influence. He proposed a theological and eschatological metaphysics called retroactive ontology, utilizing the indeterministic interpretation of the uncertainty principle, one of the astonishing discoveries of quantum mechanics.[1] His new ontology elucidates the phenomenon whereby events, which are fully realizable solely at the eschaton, occur beforehand in history. They are God's influences on the world, and the resurrection of Jesus Christ stands as a quintessential manifestation of such an event. His metaphysics harmonizes with not only the influences of God, but also the historical facticity of the resurrection of Jesus.

In this paper, I explore how his systematic approach sought to rescue theology from the brink of academic crisis and reinstate it as a vital, intellectually viable field of study. The first section begins with Pannenberg's definition of the task of theology. In Section 2, I describe how he showed that theology is the science of God. Then, I discuss several characteristic terms in his writing, such as history, facticity, the historicity of resurrection,

and prolepsis. The subsequent section explores his theology of nature, and his ontology follows thereafter.

## 2. Task of Theology

Pannenberg's purpose in doing theology was to understand God rationally in faith as much as possible, and to share this understanding in order to persuade people to seek God, have hope, and follow God's will. Of the task of theology, Pannenberg said in his later years that the doctrine of God "can only be done in the form of a systematic theology, a coherent account of how the world and especially human nature and history are related to God as creative source and ultimate destination of all things. A Christian systematic theology has to deal with the task in the form of a history of the world and of the human race, a history that accomplishes the intrinsic aim of the act of creation and overcomes the failures and shortcomings of the creatures in order to fully realize the kingdom of the creator in the world of his creatures" (Nelson 2013, p. 160). This utterance shows that his viewpoint had not changed much from the time when he published his series of *Basic Questions in Theology: Collected Essays*. In volume 1 of this series, the German theologian argued that if God is the creator of all things, the task of theology should be "to understand all being [*alles Seienden*] in relation to God" (Pannenberg 1972a, p. 1). Moreover, he stressed that to carry out this task, theology had a duty to relate all truth and thus "all the knowledge of the extra-theological sciences" to God. However, this operation should be carried out rationally in faith. For Pannenberg, both Schleiermacher's reliance on subjectivity and Barth's presumption of the truth of Christianity were no longer plausible. Therefore, in volume 2, he emphasized the imperative need for "a rational account of the truth of faith" (Pannenberg 1972b, p. 53). This thought led him to assert that theology should be a science, that it should employ a similar methodology to other sciences, and thereby examine its truth claims as if theology were a natural science.

## 3. Theology as Science

Wissenshaft, from the German word wissenshaftstheorie—which was translated as "the philosophy of science" in the book *Theology and the Philosophy of Science* (Pannenberg 1976)—has a broader meaning in German than just "science". In English, science is usually understood as the natural sciences, but the term wissenshaft in German includes all scholarly disciplines. In this book, Pannenberg wrote that the object of the philosophy of science is "to reach a new self-understanding of science in general which will provide the basis for a new ordering of scientific disciplines and their method" (Pannenberg 1976, p. 4), and that theology should not ignore the demand that it be a science. For Pannenberg, it is important that theology is a science and is acknowledged as a science by other sciences. To establish this, he first needed to show what science is by tracing the history of discussions on the subject.

Although logical positivism—which asserts that only statements verifiable by empirical observations are cognitively meaningful—excludes the influential power of God from metaphysics, its limitations have been revealed by Karl Popper. Popper insisted that the principle of induction cannot confirm the certainty of a theory because it can be proven false even if a single counter-example is found. Until all things are examined, the theory remains provisional. However, since it is impossible to examine everything in the universe, "these theories never satisfy the idea of truth as correspondence definitively or without qualification, but are at best only provisionally established" (ibid., p. 41). However, Popper did not consider theology a science because its propositions are not falsifiable, and thus its statements are not scientific statements, even though he did admit theology was meaningful. As Thomas Kuhn saw, however, the observations made are determined by the observers' paradigm. Moreover, Kuhn showed that "even in the natural sciences the testing of hypothetical laws usually takes the form not of direct attempts to falsify them, but more often of a comparison of 'the ability of different theories to explain the evidence at hand'" (Pannenberg 1976, p. 66). This method opens up the way to legitimizing the

use of hermeneutical methods in the field of the human sciences and the conclusion that every methodology is in fact hermeneutical because the meaning is always determined by the context.

### 3.1. Hermeneutics

After showing that even in natural science meaning is not free from the observer's anticipation and context, Pannenberg asserted that we should not exclude "philosophy (or metaphysics) from the class of scientifically meaningful statements" (ibid., p. 68) if we apply the idea of critical examination. For the German thinker, philosophical assertions are in fact always hermeneutical because they "are always about reality as a whole, whether it is all the aspects of a single phenomenon or the whole of reality as the semantic context of every individual phenomenon" (ibid., p. 68). Our experience of reality at any moment is incomplete because reality is still in process, however. That is why philosophical interpretations of reality as a whole are always provisional and can be treated as hypotheses. These hypotheses can be tested and should be tested, but this is difficult because "the incompleteness of experience and the nature of reality as a continuing process means not only that new individual cases may turn up—as in the case of induction—but also that they may shift the total structure of events into a new perspective" (ibid., p. 69).

Hermeneutics finds meaning in specific instances from the anticipatory relation to the whole. The aim of hermeneutics, Pannenberg wrote, "is the understanding of meaning, and meaning is to be understood in this context as the relation of parts to whole within a structure of life or experience" (ibid., p. 156). For Pannenberg, every system includes the relation between the parts and the whole, no matter what the system is. From Dilthey's work on the contextual concept of meaning, Pannenberg derived a concept of meaning that "depends on a final, all-embracing totality of meaning in which all individual meanings are linked to form a semantic whole". These thoughts probably led him to see revelation as history, like the title of his famous book, in terms of which it is history which provides information about God, the all-determining reality.

### 3.2. Theology as the Science of God

Since every scientific meaning is interpreted in context, if theology is science, it is also hermeneutics. If theology is hermeneutics, theology must be concerned with the contextual whole, the totality of reality. Furthermore, if God is construed as the all-determining reality,[2] the concept of God should correlate with the concept of the totality of reality, because "everything must be shown to be determined by this reality and to be ultimately unintelligible without it" (ibid., p. 302). Therefore, theology, which studies the idea of God, is a hermeneutics which concerns the contextual whole and thus theology is science, and specifically, the science of God.

The problem here is that "statements about God obviously cannot be verified on their object" ((ibid., p. 331), although science should be examined. This is because, if the all-determining reality correlated with the totality of reality could "be measured like any readily reproducible finite entity" (ibid., p. 332), it would be contradictory to its definition. This means that theological assertions about God and His actions in history "cannot be directly verified against their object" (ibid.). However, there is a roundabout way to test the assertions. They can be tested by their implications for the understanding of finite reality. Like other sciences, theology can also have a framework of theoretical networks, and theological statements "can be verified only by reference to their function in the system of theoretical formulation" (ibid.). Of course, these statements have the form of hypotheses to be tested.

Pannenberg's definition of "theology as the science of God" and as "the study of the totality of the real from the point of view of the reality which ultimately determines it both as a whole and in its parts" (ibid., p. 303) is therefore acceptable. Also, everything given to human experience in the history of the universe can be the object of theology as an indirect

way of understanding God. This notion is related to another of Pannenberg's theological notions: revelation as history.

## 4. Revelation as History

In 1961, along with his colleagues, Pannenberg presented his new thesis, "revelation as history", and created a sensation by refuting the theology of the Word of God, as delivered by Karl Barth and Rudolf Bultmann. The main point of the new theory was that the whole of history is the revelation of God (while others thought that only the Word of God is the revelation of God). However, the Munich thinker argued that revelation is neither hidden nor direct. Instead, it is indirect and open to all people in universal history. In criticizing the theology of the Word of God and to support his argument, he analyzed the concept of revelation and humans' awareness of revelation in various ways.

1.  "If God is already totally revealed in the special decisiveness of the Christ event, then he cannot in consistency be 'also' revealed in other events, situations, and persons" (Pannenberg et al. 1968, p. 6). In other words, a multiplicity of revelation is rejected. Moreover, if there is a special form of true revelation, this cannot be a veiling because if God reveals Himself he does not veil Himself.
2.  Pannenberg pointed out that there is no foundation in the Bible for the concept of self-revelation. God's revelation has been always understood in indirect ways, such as through prophets, laws, and the word of God, and these have been proven, not by God's manifestations, but by God's actions.
3.  If we believe that God is the author of everything and "wish to understand the indirect self-communication that resides in every individual act of God as revelation" (ibid., p. 16), "then we can view the various events of revelation as component parts of the one all-embracing event of self-revelation to which each of them makes its own specific contribution" (Pannenberg 1991, p. 244). "Only then is it possible to understand the totality of God's action—and if God is one then that means everything that happens—as his revelation" (Pannenberg et al. 1968, p. 16).
4.  According to Pannenberg, however, "revelation is not comprehended completely in the beginning, but at the end of the revealing history" (Pannenberg 1976, p. 131). Since individual events are components of the whole of history, none of them can fully reveal God. This finitude of indirect revelation is grounds for placing revelation at the end of history. Only at the end of history is history itself considered as the whole of history. Only then is God completely revealed. God's deity is proven by His actions, showing His lordship over creation.

## 5. Role of the Resurrection of Jesus and Its Historicity

In the history of Israel, before the time of the exile, God's divine status was demonstrated by His power over creation, through miracles, the exodus from Egypt, and the possession of the land of Canaan, thereby prompting Israelites to put their faith in Him. Only God's rule over creation can prove His deity and this deity will only be fully revealed at the eschaton. This raises a question, however. How can people in the process of history know God as One, i.e., not just God for the Israelites, but for the whole creation, and believe in Him before the eschaton? The answer, Pannenberg thought, lay in the life of Jesus of Nazareth and his proclamations.

Pannenberg argued that the universal revelation of the deity of God is realized "first in the fate of Jesus of Nazareth, insofar as the end of all events is anticipated in his fate" (ibid., p. 139). In his life, the man Jesus always preached the kingdom of God, but the coming of God's rule was not a political revolution as his contemporaries expected. Rather, he spoke about the need for our total commitment to God. We should seek first God's kingdom and God's righteousness, as written in Matthew 6:33, and love God with all our heart, and with all our soul, and with all our might, as written in Deuteronomy 6:4. God already comes with His rule to those who open themselves to this summons. "The particular dynamic

of Jesus' message of the *basileia*, then, is that the rule of God is imminent but that it also emerges from its futurity as present" (Pannenberg 1994, p. 330).

Jesus of Nazareth, as the son of God, preached who God is and what God wants us to do. Jesus himself proclaimed that he is the son of God. Yet only subordination to the rule of the one God that he himself preached could prove that he truly was the son of God (ibid., p. 373). Jesus showed what total commitment to God is through his relationship to God and by his obedience which extended to being crucified. The resurrection of Jesus by the Spirit showed that God justified the condemned and executed Jesus and confirmed his messages (ibid., p. 344). For Pannenberg, even Jesus' baptism or the origin of his earthly life as the birth of the Son of God can only be related to the divine sonship of Jesus, "in the light of the Easter event and as an expression of its confirmatory function" (ibid., p. 366). Therefore, the resurrection of Jesus proved his deity as the Son of God.[3] "The resurrection of Jesus is the basis of Christian faith, yet not as an isolated event, but in its reference back to the earthly sending of Jesus and his death on the cross" (ibid., p. 344). This is why the facticity of the resurrection of Jesus of Nazareth has crucial importance in Pannenberg's theology.

### 6. Facticity of the Resurrection of Jesus of Nazareth

Pannenberg emphasized the facticity of the Easter event because all discussion of its meaning would be a waste of time if the event had not actually taken place (ibid., p. 346). First, the German theologian offered as ground for the facticity of the resurrection "the appearances of the risen Lord to his disciples, along with the discovery of the empty tomb of Jesus in Jerusalem" (ibid., p. 353). Of course, since these testimonies could be lies, in examining them, he did not rely solely on the authority of the Bible. The fact that Paul's own resolute appeal in Galatians 1:1, 12 was accepted implies that the appearance of the risen Lord to the disciples had actually taken place before Paul met Jesus (ibid., p. 355). Otherwise, his appeal might not have been accepted because nobody would have believed his testimony about the appearance of Jesus. Pannenberg also stressed the importance of the discovery of the empty tomb as a ground for the historicity of the Easter event because it supports the claim that the appearances of the risen Lord are not mere hallucinations, but true realities, even though "primitive Christian conviction as to the resurrection of Jesus rests not on the finding of his empty tomb but on the appearances" (ibid., p. 359).

It is true that these grounds do not prove the historicity of the Easter event completely and are not easily accepted by many because of the unusualness of the event. However, Pannenberg contended that "historicity does not necessarily mean that what is said to have taken place historically must be like other known events" (ibid., p. 360). The normality of an event may play a role in the validation of a certain claim, but it is not a strict condition of the actual facticity of that claim. "Hence a reference to the otherness of the eschatological reality of resurrection life compared to the reality of this passing world does not affect the claim to historicity that is implied in the assertion of the facticity of an event that took place at a specific time" (ibid., p. 361).

According to Pannenberg, the evaluation of the facticity of the resurrection of Jesus of Nazareth "depends not only on examining the individual data (and the related reconstruction of the event), but also on our understanding of reality, of what we regard as possible or impossible prior to any evaluation of the details".[4] There is therefore no reason to surrender to secular criticism and forsake the historicity of the Easter event.

### 7. Eschatological Prolepsis

The concept of prolepsis is essential in Pannenberg's theology, according to Ted Peters. For Pannenberg, a prolepsis is "a concrete pre-actualization of a still outstanding future reality. It is eschatological reality appearing within history ahead of time" (Peters 2014, p. 380). The resurrection of Jesus of Nazareth is the focal prolepsis which proves God's deity because it shows who Jesus is and approves what Jesus said. From the life and proclamations of Jesus, we can proleptically see the divine rule of the kingdom of God. Through Jesus, God's deity was proleptically proven. We can therefore anticipate the divine

rule at the eschaton. However, the crucial role of Jesus' resurrection is not only proving his and God's divinity, but also gives hope of the general resurrection. "He is 'the first fruits of those that have fallen asleep' (l Cor. 15:20), 'the first-born among many brethren' (Rom. 8:29), 'the first-born from the dead' (Col. 1:18; cf. Rev. 1:5), the initiator of a new life (Acts 3:15)" (Pannenberg 1994, p. 348).

We can anticipate the new life at the eschaton from the relation of Jesus of Nazareth to God because he showed it proleptically in his life and proclamation, and "the eschaton has in some sense begun in Jesus' resurrection" (Clayton 1988, p. 131). By this proleptic event, God showed His confirmation of the claim "that the imminent rule of God that Jesus proclaimed was about to break in, and in fact was already doing so for those who trusted his message" (Pannenberg 1994, p. 345). From this prolepsis, we can anticipate that "believers receive a share in the sonship of Jesus Christ," by the Spirit (ibid., p. 317). On the ground of Jesus' proleptic resurrection, Pannenberg argued that the power of the future determines the meaning of the present in the anticipation of the eschatological kingdom of God. The future of God "is constitutive for what we now are and already have been" (Pannenberg 2009, p. 551). Therefore, prolepsis involves retroactive causation. This explains why Pannenberg's theology is necessarily eschatological. His eschatological theology necessitates a retrospective ontology, in which he theologically interprets and applies the discoveries of contemporary physics, engaging in dialogue with natural science.

## 8. Toward a Theology of Nature

The origin of "natural theology" is found in Panaetius, who "used the term for the philosophical doctrine of God as distinct from the mythical theology of the poets on the one side, and on the other the political theology of the cults which the states set up and supported" (Pannenberg 1991, p. 76). The philosophical knowledge of God is "natural" not because of its suitability to human nature or human ability of understanding but because of its correspondence to "the nature of the divine or the truth of God" (ibid., p. 77). The proof of the existence of God, which was later criticized by philosophers and as a result abandoned by many theologians, was not studied in early natural theology. Since the existence of God was an indisputable premise, natural theology focused on what God's nature was (ibid., p. 78). "The natural theology of the philosophers had formulated a criterion for judging whether any God could be seriously considered as the author of the whole cosmos" (ibid., p. 79). To proclaim God as the Creator of this universe and as the true God of all human beings who saves them in Jesus Christ, theology therefore had to engage with this criterion. Theologians had to prove either that the God of Christianity met this criterion or that this criterion was not proper in describing the attributes of the Creator (ibid., p. 80). The notion of natural theology was damaged when it came to be considered the opposite of revealed theology, as was the case in early Protestant theology. "'Natural' no longer meant 'in accordance with the nature of God' but 'in accordance with human nature'" (ibid., p. 81). Ritschl, who wanted freedom from the connection to metaphysics in scientific positivism, criticized philosophical theology, and this criticism was developed further by Karl Barth (ibid., p. 100). However, we would lose the foundation of the facticity of every statement about God if we forsook any possibility of natural knowledge of God. Natural theology helps to carry out the task of theology in that Christianity must explain its proclamation about God plausibly for every age.

In this regard, as Ted Peters has pointed out, Pannenberg rejected the two-language approach that is so problematic in the dialogue between theology and natural science. "According to the two-language theory, scientists and theologians work in separate domains of knowledge, speak separate languages, and, when true to their respective disciplines, avoid interfering in each other's work" (Pannenberg 1993, p. 4). Instead, the Munich theologian contended that the two disciplines should talk to each other, and influence each other, pursuing a theology of nature (namely, a theology that "relies on both modern science and classical Christian commitments regarding creation, conservation, and governance" (Pannenberg 1993, p. 2)). This is because, as mentioned earlier, if theology is the science

of God, and if God is the all-determining reality, then the nature given to us cannot be unrelated to God. A God who cannot be the origin and perfecter of this nature cannot be the power that determines all the reality of being. Consequently, such a God cannot be the true God. If theology is to contemplate the deity of God, then theology must conceive of God as the power that determines nature as well as human history (ibid., p. 75). Therefore, a theology of nature would have to present nature as a creature with its present condition and all the processes that led up to it, including its beginning history. "It would have to relate all of nature to the reality that is the true theme of theology-the reality of God" (ibid., p. 73).

Also, since Pannenberg thought that scientific theories were not yet perfect, he criticized the scientific vision of nature as incomplete without the idea of God. He emphasized that scientists should adopt the idea of God to their world view because scientific theories cannot explain fully the reality of the world without God, the Creator (ibid., p. 16). From this perspective, Pannenberg addressed five questions to scientists (ibid., pp. 17–19), questions which involve the contingency and regularity found in nature, time and eternity, and the self-giving dynamics of the Spirit (Pannenberg 2008, pp. 30–39). He thought that the irreversibility and temporality of nature implied the ontological contingency of nature. Thus, the regularity of the natural law on which deists and atheists rely is in fact sustained by the power of God. It is thus not so strange that many theistic philosophers and scientists accepted the principle of inertia as divine faithfulness at the time when it was first postulated. Furthermore, when it comes to the concept of the space-time continuum in physics, we need to consider the relation of time to eternity[5], Pannenberg argued. In these points, the idea of God would support science in broadening its viewpoint and transcending its finitude. Moreover, the concept of the self-giving power of the Spirit as the force field will help all of us understand how God acts in nature as the all-determining power of the future. This thought connects with Pannenberg's notion of metaphysics.

## 9. Retroactive Ontology

Pannenberg presented a new definition of the concept of substance by arguing that things will "be what they are, substances, retroactively from the outcome of their becoming on the one hand, and on the other in the sense of anticipating the completion of their process of becoming, their history" (Pannenberg 1990, p. 107). He wanted to overcome the limitations of atomism because it leads us to "fall into the mode of thinking of materialism".[6] For Pannenberg, a proper ontology must provide an explanation of being and time. Further, this ontology should consider part and whole, the eschatological totality, because the "meaning of the events and things that we experience changes with the alteration of the context over the course of time" (ibid., p. 104). He believed that theology can contribute to building such a new ontology, for, "according to Christian doctrine, God as Creator is already related to each of his creatures in love; all created life is to be understood as a form of participation in the divine eternity, however weak or limited this participation may be. The length of time granted to each creature can then be interpreted as an anticipation of the final completion that is expected from the future of God's rule. At that time, creatures, despite their finitude, will participate in the eternity of God, at least to the extent that they do not close themselves off from that eternity" (ibid., p. 97).

This was revealed in Jesus' work and proclamations and was proven by his proleptical resurrection. The resurrection of Jesus of Nazareth is therefore the important ground for Pannenberg's idea of retroactive ontology. Although Pannenberg himself wrote that he has no tendency to "any particular philosophical system," even his own, as Ted Peters pointed out, "a consistent and coherent structure of underlying reality can be discerned in his work" (Peters 2014, p. 371). That is, a retroactive ontology derived from Pannenberg's notion of hermeneutics can be identified, which relies on the totality, and on the understanding of God as the all-determining power of the future and of the proleptic event of Easter. Pannenberg believed that this ontology was a ground for indeterminism and could liberate the future from its strict causal relation to the past.

### 9.1. Contingency and Quantum Indeterminacy

Until the nineteenth century, "a sort of religious faith, a late form of Greek cosmos piety, played a role in the deterministic worldview of classical natural science" (Pannenberg 1993, p. 77). Pannenberg believed that the illusion of such a determinism was shattered by contemporary physics, and quantum mechanics, because he followed the Copenhagen interpretation of quantum mechanics.[7] Pannenberg also used this indeterminist interpretation of quantum physics to support his concept of contingency. Moreover, he suggested to extend the applicable scope of ontological uncertainty derived from the uncertainty principle of Werner Karl Heisenberg from the micro level to the macro level (at least to high molecular structures) (Pannenberg 1993, p. 116, see endnote 11).

However, Pannenberg was careful beforehand not to misunderstand and misuse the discovery of the contingent phenomena of quantum mechanics and thermodynamics. According to the German theologian, if we regard the phenomena as contingent because they are exceptional cases from other natural laws, then we "fall victim to the fatal mistake of seeing God at work precisely in these gaps, so that every scientific advance would be a further blow to theology" (Pannenberg 1994, p. 71). Rather, we should consider even regularity, which is regarded as natural law, as a contingent positing of God the Creator. Without such a conceptual integration with contingency, the truth of the belief in creation will fall into the trap of deism or atheism.

Nevertheless, following Hans-Peter Dürr's argument on the concept of possibility in quantum indeterminacy, Pannenberg asserted that "if we view the occurrence of microevents at each present moment as a manifestation of the future (deriving from the possibility field of future events)" (ibid., p. 100), "a meaningful connection might then be seen between the statement that extrapolation into the future is not possible and the statement that this world in some sense takes place afresh every moment" (ibid., p. 99). On the ground of quantum indeterminacy, Pannenberg therefore contended that every event in nature is ontologically contingent, and we can thus expect the nature's openness to the future.

### 9.2. Spirit as Field

Pannenberg rejected atomism because he thought that in metaphysics, atomism runs the risk of falling into materialism. However, Pannenberg also knew that, in order to suggest a new ontology, he had to offer an explanation of "the relations between this sort of philosophical description and the natural-scientific description of material processes" (Pannenberg 1990, p. 108). To achieve this task, the German thinker employed the electromagnetic field theories of Michael Faraday, who had suggested a concept of "a single field of force that determines all the changes in the natural universe" (Pannenberg 1993, p. 38), and "that embraces one or more bodies" (Pannenberg 1994, p. 81). In Faraday's view, "the material particle is the point of convergence of lines of force, or a cluster of such lines over a given period" (ibid.). By borrowing this concept, Pannenberg could explain what the material particles and the force between them were.

The concept of the Spirit as field was originally discussed in the Stoic school. The Stoics followed the doctrine of the divine *pneuma,* which "permeates all things, that holds all things in the cosmos together by its tension (*tonos*), and that gives rise to the different qualities and movements of things" (ibid.). This concept presented a problem that early Christians could not accept, however, because the Stoics "conceived of it as a subtle material element" (Pannenberg 1993, p. 39). Since the early Christians thought of the divine reality as purely spiritual, they rejected this idea. However, in the field concept of modern physics, this difficulty was no longer troublesome because the need for ether as a substratum was banished by Albert Einstein's Special Relativity. Instead, Pannenberg believed this concept of the field corresponded to the older doctrines. For him, this "seems to offer a modern language that possibly can express the biblical idea of the divine spirit as the power of life that transcends the living organism and at the same time is intimately present in the individual" (Pannenberg 1993, p. 24). This is because he thought that this scientific field

concept aligns more closely with the scriptural portrayal of God as spirit, avoiding the complexities associated with the concept of God with bodily existence. Also, he contended that it is reasonable to "relate the field theories of modern physics to the Christian doctrine of the dynamic work of the divine Spirit in creation" (Pannenberg 1994, p. 82). Since the Spirit is the Spirit of life and freedom, the concept of the Spirit as a field, which permeates all beings, and creates and preserves them, could be one of the grounds for the idea that all events in nature are the results of the free acts of God, the all-determining power of the future. For the future is the field of the potentiality, and thus "the basis of the openness of creation to a higher consummation and the source of what is new, i.e., of contingency in each new event" (ibid., p. 97). Then, "the reality of God in terms of the comprehensive field of eternity, comprising time and space and dynamically producing the temporal existence of creatures in space through its futurity in relation to all potential events" (Pannenberg 1992, p. 307).

## 10. Conclusions

In concluding our exploration of Pannenberg's method and metaphysics, it is crucial to encapsulate the distinctive features that define his theological approach. Notably, Pannenberg's methodology is hermeneutics that considers the totality of the history of the universe as the contextual whole. Thus, his theological method is historical in nature, acknowledging that the full revelation of God unfolds progressively through the expanse of time. It is eschatological, pointing towards the culmination of history where ultimate truth and divine purpose are fully realized. His method underscores the importance of reason and rationality, arguing for a theology that is coherent and intellectually defensible. Pannenberg does not restrict his theological sources to sacred texts alone; instead, he regards the totality of history—every event that has occurred—as the canvas for divine action and revelation. This holistic use of historical data underpins his systematic theology, affirming that every facet of the history of the universe contributes to our understanding of God and His relationship with the world.

Pannenberg thought the resurrection of Jesus as the origin of Christian faith and sought to gain academic recognition for theology; therefore, a metaphysics that could make this possible was necessary. In fact, to rationally discuss not only the resurrection but also God's influence within the world, a metaphysics capable of encompassing this is needed. In the age of science, a metaphysics that harmonizes with both science and theology is especially required to discuss the doctrine of God. If this new metaphysics can make God's activity explainable, then a Scriptural interpretation based on this metaphysics will allow for an objective acceptance of God's actions within the Bible. (If someone interprets God's actions in the Bible only subjectively, it is because their metaphysics, whether they are aware of it or not, cannot accommodate it.) Pannenberg's metaphysics posited the ontological contingency of the world based on the indeterministic interpretation of the principle of uncertainty in quantum mechanics, and he argued that in a world of contingency, the actions of God do not conflict with the laws of nature. Within this metaphysical framework, the actions of God, including the resurrection of Jesus presented in the Bible, could be accepted as objective historical facts.

Pannenberg's methodical approach to theology presents a compelling case for the integration of faith with intellectual rigor. His work, deeply embedded in a commitment to truth and clarity, challenges the dichotomy between sacred and secular knowledge. Pannenberg's systematic theology, as explored in the main body of this paper, does not shy away from the critical scrutiny of modern thought; instead, it invites it, using it as a forge for a more resilient understanding of Christian doctrine. The conclusion of this study affirms that Pannenberg's theological contributions are not only significant in their academic merit but also in their ability to provide a credible voice for faith in a scientific and philosophical landscape.[8] His legacy endures in the way contemporary theology can engage with the world and uphold the essence of the Christian faith.

Nonetheless, while Pannenberg's methodology has offered a robust framework for dialogue between faith and reason, it is not without its critiques. Some argue that his systematic approach may at times seem too optimistic in its endeavor to reconcile faith with the empirical demands of scientific scrutiny. Critics have pointed out that by seeking to explain too much within the confines of rationality, there is a risk of constraining the transcendent and mysterious nature of the divine. Also, questions are raised about the practical implications of Pannenberg's theology: Does it remain too abstract and theoretical to resonate with the experiential dimensions of faith for many believers? Furthermore, as even Pannenberg conceded, certainly, rigid verification as understood by logical positivism is impossible.

This presents the toughest test for theology. However, "such a strict verification is not possible even for the posited laws of physics".[9] Although the eminent theologian of the 20th century, Karl Barth diagnosed, "[t]heological knowledge, thought, and speech cannot become general truths, and general knowledge cannot become theological truth [and thus] [d]istressing as the situation may be, there is no getting around the special character and relative solitude of theology in relation to other sciences" (Barth 1992, p. 114), we now face a challenge that theology cannot sidestep through any claims of its own supremacy or the inherently unprovable nature of divine truth. "Such protestations may immunize theological discourse against critique but, at the same time, render it absurd because propositions that are, in principle, closed to any kind of critical questioning of their truth-claims are not propositions at all and hence can no longer be seriously regarded as propositions" (Pannenberg 2008, p. 16). As Jürgen Moltmann pointed out, if Christian theology is just for church in a narrow sense, there is no place for theology in public or state universities because it does not require the universal horizon. However, "if Christian theology sees itself as a function of the kingdom of God, for which Christ came and for which the church itself is, after all, there, then it must develop as a public theology (*theologia publica*) in public life" (Moltmann 2000, p. 79). To achieve this goal, out of its own eschatological theology, Christian theology should develop a new creation theology, and out of that a new type of natural theology (ibid., p. 80), namely the theology of naure[10]. And that is what Pannenberg's theology strove for.

**Funding:** This research received no external funding.

**Data Availability Statement:** No new data were created or analyzed in this study. Data sharing is not applicable to this article.

**Conflicts of Interest:** The author declare no conflict of interest.

**Notes**

[1] (Pannenberg 1994, p. 330). Pannenberg, building upon Hans-Peter Dürr's interpretation of quantum indeterminacy, understands the fundamental contingency of events at every moment as a manifestation of the future in each moment.

[2] It is true that we cannot prove God as the all-determining reality when we talk about the idea of God. Such a metaphysical assertion cannot essentially be proven by any empirical examination. Thus, Thomas Aquinas or Paul Tillich just argued that God is the ground of Being but did not prove it.

[3] Here, we need to think more about the preexistence of the Son and his eternal relation to the Father and the Spirit that is the relation of Trinity because in Pannenberg's theology, the life of Jesus is a proleptical economic representation of the relation of the eternal immanent Trinity from the future.

[4] (Pannenberg 1994, p. 362). This phenomenon reminds us of the limitations of evaluations in the natural sciences which are not free from their paradigm as context, as discussed in relation to Pannenberg's theory of science above.

[5] According to Pannenberg, "changes in the scientific description of time and space like the space-time concept of the general theory of relativity contribute less than one might think to the philosophical question of the nature of space and time". He also contended that since "the infinite and undivided whole of space and time—or of space-time, for that matter—precedes all measurement", "the infinite space of God's immensity and the infinite whole of simultaneous presence that is God's eternity are implicated and presupposed in our human conceptions and in our measurements of space and time". Accordingly, God's eternity is distinct from the time of His creations but constitutes it, and His immensity constitutes the space of His creatures (Pannenberg 2008, p. 69).

6    (Pannenberg 2008, p. 183). Here, Pannenberg wrote that even process philosophy is atomistic when it talks about undivided actual occasions.

7    (Pannenberg 2008, p. 47). Besides quantum mechanics, Pannenberg believes that the thermodynamic investigations of fluctuations in dissipative systems have demonstrated unpredictability and contingency even in the macroscopic world, and that the observation of chaotic processes reveals that, while theoretically deterministic, they are non-deterministic at a bifurcation point.

8    Pannenberg's theological work was so remarkable that even his most formidable critics had to acknowledge that his work "is likely to prove the greatest Systematic Theology of his generation" (Schwöbel 1996, p. 497).

9    This is because "no general rule can be exhaustively tested by a finite number of cases to which it applies" (Pannenberg 2008, p. 21).

10   Here, I refer to a theology that "relies on both modern science and classical Christian commitments regarding creation, conservation, and governance." (Pannenberg 1993, p. 2). In the field of theology and science, Ian Barbour's concept of "the theology of nature" has garnered significant scholarly attention and is widely recognized as a seminal contribution to the field. According to Barbour, a theology of nature "starts from a religious tradition based on religious experience and historical revelation. But it holds that some traditional doctrines need to be reformulated in the light of current science." (Barbour 2000, p. 31). I believe that the two individuals' descriptions of "the theology of nature" are compatible with each other.

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
