# Peer review of "Wolfhart Pannenberg’s Theological Method and Metaphysics"

_religions, doi:10.3390/rel15060714_

Round 1
Reviewer 1 Report
Comments and Suggestions for Authors
The author helpfully explains Pannenberg's theological approach to rationality and science. Using retroactive ontology as it relates to viewing theology as a science is helpful. That said, this article reads more like a summary statement of Pannenberg's various approaches rather than a sustained theological argument. The framing of Pannenberg's method through the idea of logical positivism is an interesting idea. So if logical positivism can be expanded and described, then this article has the potential to provide something novel interesting as it relates to how Pannenberg's retroactive ontology specifically counters logical positivism.
The author discusses history, facticity, the historicity of resurrection, and prolepsis, but I'm not quite clear on how these things are making the case for the overall argument. The role of retroactive ontology as a counter to logical positivism would be a helpful approach to take.
I would recommend the author consult more recent scholarship. For example, also consult "God and the Future" by Christian Moostert for more on a retroactive ontology, in addition to Stanley Grenz, "Reason for Hope." Jae Yang's "Christianity Outside the Church" is helpful in terms of relating Pannenberg's view of theology to the public sphere using academic and rational ideas. Robert John Russell's "Time in Eternity" provides a helpful understanding of Pannenberg's view of science as a scientific method. As far as Pannenberg's scientific method is concerned, Kuhn and Popper are helpful utilized, however, a Lakatosian perspective provided by Nancey Murphey will be helpful as well in the discussion.
Author Response
I hope this message finds you well. I wanted to take a moment to express my sincere gratitude for the time and effort you have invested in reviewing my manuscript titled Wolfhart Pannenberg’s Theological Method and Metaphysics. Your thorough and thoughtful critique has provided me with invaluable insights and has undoubtedly contributed to the improvement of my work.
I greatly appreciate the depth and clarity of your comments, which have highlighted both the strengths and weaknesses of my manuscript. Your expertise and critical eye have helped me identify areas that require further clarification and refinement.
While I agree with many of the points you have raised, I would like to humbly address a few aspects where my perspective differs slightly.
This article is submitted to the special issue "Theological Metaphysics and Scriptural Interpretation" of Religions. The purpose of this special issue was to discuss how metaphysical presuppositions are related to biblical interpretation (especially the historical Jesus).
After the rise of logical positivism, claiming the historical facticity of Jesus' resurrection was considered unscholarly. In order for theology to assert the resurrection of Jesus, which is the core foundation of the Christian faith, while still being recognized as academic, a new metaphysics was needed. Pannenberg presented a new metaphysics for this purpose, and I believe it was somewhat successful. Moreover, I think his methodology justifies his metaphysics. The purpose of this article is to explain Pannenberg's metaphysics, which supports the historical factuality of Jesus' resurrection, and his methodology for constructing that metaphysics. Therefore, it may appear to be a summary of Pannenberg's methodology rather than an article presenting and arguing for a new idea. However, that is because it is the purpose of this article. In the newly uploaded paper, I have highlighted the relevant parts in sky blue in the introduction (lines 28-39) and conclusion (lines 425-439).
Next, I have written about Pannenberg's use of Popper's critique of logical positivism to point out the limitations of logical positivism at the beginning of Section 3. However, since Popper's falsificationism also implies the testability of hypotheses, theological statements are not recognized as scientific statements in falsificationism. Pannenberg goes beyond the limitations of Popper's falsificationism by using Kuhn's concept of paradigm shift. I have highlighted lines 79-92 of Section 3 in sky blue for this.
As you mentioned, Pannenberg was criticized for not addressing Lakatos' perspective. In response to this, Pannenberg replied as follows:
“When I wrote my book Theology and the Philosophy of Science … at that time, around 1970, the latest phase of discussion was represented by Thomas Kuhn.” “In terms of systematic structure, the framework of theological explanations as I envision it may be adequately described in Lakatosian terms … I tend to emphasize more
strongly the unity of elements in a systematic interpretation [than Lakatos].” (Wolfhart Pannenberg, “Theological appropriation of scientific understandings: response to Hefner, Wicken, Eaves, and Tipler”, Zygon, 24.2 (Jun 1989), 258-259.)
The recommended books were not directly cited in this paper, but they were of great help in broadening my perspective. I think I can add them to my bibliography, if you agree.
Once again, I am truly grateful for the time and effort you have dedicated to reviewing my manuscript. Your valuable feedback has challenged me to think more critically about my research and has provided me with a roadmap for improvement.
Thank you once again for your commitment to the peer-review process and for your contributions to the advancement of our field.

Reviewer 2 Report
Comments and Suggestions for Authors
Please see the attached for reviewer comments.

Generally quite good. Slight editing recommended before publication.
Author Response
Thank you so much for your valuable suggestions for revision. Please note that I have responded to each of your comments. (As per your suggestions, the revised parts of the attached paper are highlighted in yellow.)
I think this is an excellent paper. I hope to be in contact with you following its publication.
Here are a couple of suggestions for improving the article:
Line 3-4
Abstract: After the rise of logical positivism, even in the realm of theology, there was a trend to give up on objectively accepting the actions of God in history and to create an atmosphere of separating faith and reason
Suggested Abstract: After the rise of logical positivism, even in the realm of theology, there was a trend to give up on accepting the actions of God in history as objective acts and to create an atmosphere of separating faith and reason
Thank you I applied your suggestion to my paper
Line 31
utilizing the indeterministic interpretation of the uncertainty principle, one of the astonishing discoveries of quantum mechanics
Author: Please include a reference to his use of quantum indeterminism here at the outset of your paper.
I added a footnote 1 as you suggested.
Pannenberg, building upon Hans-Peter Dürr's interpretation of quantum indeterminacy, understands the fundamental contingency of events at every moment as a manifestation of the future in each moment.
Accordingly, footnote 26 has been changed.
Wolfhart Pannenberg, Systematic Theology, trans. Geoffrey W. Bromiley, vol. 2 (Grand Rapids, MI: Eerdmans, 1994), 330.
-> Pannenberg, Systematic Theology, 2:330.
Line 130
although science should be examined
Author: What does that mean?
This is because Pannenberg desired theological studies to be recognized as academically legitimate within academia. He acknowledged that doctrinal statements in theology hold the status of hypotheses in the perspective of philosophy of science, and that their truth can be questioned.(Systematic Theology 1:56) Furthermore, he argued that when theologians make propositions, they must clarify them in all aspects relevant to the specific topic to be verifiable. (Theology and the Philosophy of Science, 331) However, Pannenberg holds that “theological statements cannot be tested by experiment because they do not affirm rules concerning recurring and repeatable sequences of events”, but they can be indirectly confirmed. “The test of theological affirmations is rather in their hermeneutic correctness and systematic presentation.” (The Historicity of Nature, 9)
Line 213-214
That Paul's own resolute appeal in Galatians 1:1, 12 was accepted implies that the appearance of the risen Lord to the disciples had actually taken place before Paul met Jesus
Author: Grammar?
I added “The fact” in front of the sentence.
Line 260
8. Toward a Theology of Nature
Author: It would help the reader if there were an introductory couple of sentences telling us why we transitioned to this new topic and why it's relevant to the essay as a whole.
I added the following sentence.
His eschatological theology necessitates a retrospective ontology, in which he theologically interprets and applies the discoveries of contemporary physics, engaging in dialogue with natural science.
Line 348
determinism was shattered by contemporary physics, and quantum mechanics, because
Author: Why "and" here? Normally I would say something like this: "quantum mechanics is usually interpreted philosophically as pointing to ontological indeterminism at the atomic and nuclear levels of the world". Does your "and" mean that there are other theories in physics which also suggest ontological indeterminism (and not just epistemic uncertainty, such as chaos and complexity theories)?
I added the following content to foot note 66.
Pannenberg believes that the thermodynamic investigations of fluctuations in dissipative systems have demonstrated unpredictability and contingency even in the macroscopic world, and that the observation of chaotic processes reveals that while theoretically deterministic, they are non-deterministic at a bifurcation point.
Line 370
every event in nature is ontologically contingent, and we can thus expect the nature's openness to the future.
Author: Perhaps this point is what you were anticipating in the previous quote. OK, but I'd still rewrite the previous quote which by itself and the "and" in it hard to understand.
I think now this is okay.
Line 389
However, in Faraday's concept of the field, this difficulty was no longer troublesome because the need for ether as a substratum was banished.
Author: You should probably add a sentence acknowledging that it was Einstein's Special Relativity, and not Maxwell's electromagnetism, that undercut the need for the 'ether'.
I edited as you suggested. See line 392. Thank you.
However, in the field concept of modern physics, this difficulty was no longer troublesome because the need for ether as a substratum was banished by Albert Einstein’s Special Relativity
Line 473
and out of that a new type of natural theology, namely the theology of nature.
Author: Most of us working in theology and science adopt Ian Barbour's definition of "the theology of nature" as starting with science and interpreting it theologically. So for example, t=0 in Big Bang Cosmology provides indirect evidence for a theology of creation ex nihilo. I suggest you might want to rephrase this last bit of your splendid article.
I added footnote 87 to explain it.
Here I refer a theology that “relies on both modern science and classical Christian commitments re-garding creation, conservation, and governance.” Pannenberg, Toward a Theology of Nature, 2. In the field of theology and science, Ian Barbour’s concept of "the theology of nature" has garnered significant scholarly attention and is widely recognized as a seminal contribution to the field. According Barbour, a theology of nature “starts from a religious tradition based on religious experience and historical revelation. But it holds that some traditional doctrines need to be reformulated in the light of current science.” Ian Barbour, When Science Meets Religion: Enemies, Strangers, or Partners? (HarperOne, 2000), 31. I believe that the two individuals’ descriptions of “the theology of nature” are compatible with each other.
Meanwhile, I recommend your considering the following article for inclusion here:
Robert John Russell, "Contingency in Physics and Cosmology: A Critique of the Theology of Wolfhart Pannenberg," Zygon 23.1 (March 1988).
"Theological Questions to Scientists" by Pannenberg in The Sciences and Theology in the Twentieth Century, edited by A. R. Peacocke (Notre Dame, Indiana: University of Notre Dame Press: 1981
I added them to my bibliography. Thank you so much for your valuable comments. I believe your comments have helped me enhance the quality of my paper.

Reviewer 3 Report
Comments and Suggestions for Authors
This survey is comprehensive. Good. Coherent.
Shortening some long sentences might make for easier reading.
Author Response
I wanted to take a moment to express my heartfelt gratitude for your thoughtful and positive review of my manuscript titled "Wolfhart Pannenberg’s Theological Method and Metaphysics." Your words of praise and encouragement mean a great deal to me and have further motivated me to continue my research in this field.
I am truly honored and humbled by your recognition of the value and significance of my work. Your affirmation of the manuscript's strengths and its potential contribution to the field is incredibly reassuring and validates the effort I have put into this research.
It is reviewers like you who make the peer-review process a rewarding and constructive experience for authors. Your support and appreciation for my work have not only boosted my confidence but have also reinforced my dedication to producing high-quality research.
Thank you once again for taking the time to review my manuscript and for providing such positive feedback. Your kind words will undoubtedly stay with me as I continue my academic journey.
Round 2
Reviewer 1 Report
Comments and Suggestions for Authors
Thank you for your resubmission and engaging the sources I recommended. I approve your article for publication. That said, perhaps a little more engagement with critiques of Pannenberg's new ontology would be in order as well as more engagement wit what exactly a futurist ontology might be.